# Effective degradation of various bacterial toxins using ozone ultrafine bubble water

**Fumio Takizawa**[1,2], **Hisanori Domon**[1,3], **Satoru Hirayama**[1], **Toshihito Isono**[1], **Karin Sasagawa**[1,2], **Daisuke Yonezawa**[4], **Akiomi Ushida**[5], **Satomi Tsutsuura**[6], **Tomohiro Miyoshi**[7], **Hitomi Mimuro**[7], **Akihiro Yoshida**[8], **Koichi Tabeta**[2], **Yutaka Terao**[1,3]*

1 Division of Microbiology and Infectious Diseases, Niigata University Graduate School of Medical and Dental Sciences, Niigata, Japan, 2 Division of Periodontology, Niigata University Graduate School of Medical and Dental Sciences, Niigata, Japan, 3 Center for Advanced Oral Science, Niigata University Graduate School of Medical and Dental Sciences, Niigata, Japan, 4 Division of Oral Science for Health Promotion, Niigata University, Niigata, Japan, 5 Institute of Science and Technology, Niigata University, Niigata, Japan, 6 Faculty of Agriculture, Niigata University, Niigata, Japan, 7 Division of Genome-Wide Infectious Diseases, Research Center for GLOBAL and LOCAL Infectious Disease, Oita University, Oita, Japan, 8 Department of Oral Microbiology, Matsumoto Dental University, Nagano, Japan

* terao@dent.niigata-u.ac.jp

**Data Availability Statement:** All relevant data are within the manuscript and its Supporting Information files.

**Funding:** This study was supported by Terumo Life Science Foundation (grant no. 22-IIII003 to YT),

## Abstract

Infectious and foodborne diseases pose significant global threats, with devastating consequences in low- and middle-income countries. Ozone, derived from atmospheric oxygen, exerts antimicrobial effects against various microorganisms, and degrades fungal toxins, which were initially recognized in the healthcare and food industries. However, highly concentrated ozone gas can be detrimental to human health. In addition, ozonated water is unstable and has a short half-life. Therefore, ultrafine-bubble technology is expected to overcome these issues. Ultrafine bubbles, which are nanoscale entitles that exist in water for considerable durations, have previously demonstrated bactericidal effects against various bacterial species, including antibiotic-resistant strains. This present study investigated the effects of ozone ultrafine bubble water (OUFBW) on various bacterial toxins. This study revealed that OUFBW treatment abolished the toxicity of pneumolysin, a pneumococcal pore-forming toxin, and leukotoxin, a toxin that causes leukocyte injury. Silver staining confirmed the degradation of pneumolysin, leukotoxin, and staphylococcal enterotoxin A, which are potent gastrointestinal toxins, following OUFB treatment. In addition, OUFBW treatment significantly inhibited NF-κB activation by Pam3CSK4, a synthetic triacylated lipopeptide that activates Toll-like receptor 2. Additionally, OUFBW exerted bactericidal activity against *Staphylococcus aureus*, including an antibiotic-resistant strain, without displaying significant toxicity toward human neutrophils or erythrocytes. These results suggest that OUFBW not only sterilizes bacteria but also degrades bacterial toxins.

## Introduction

A million infection-related deaths caused by bacterial pathogens occurred worldwide in 2019, with *Staphylococcus aureus* and *Streptococcus pneumoniae* accounting for approximately 26%

the Japan Society for the Promotion of Science KAKENHI (grant nos. JP20H03858, JP22KI96l4, and JP23H00445 to YT; JP20K09903 and JP23K18355 to HD; JP22K09923 to SH; and JP19H03829 to KT), and JST, the establishment of University fellowships towards the creation of science technology innovation (grant no. JPMJFS2II4 to FT). The funders had no role in study design, data collection and interpretation, or the decision to submit the work for publication.

**Competing interests:** The authors have declared that no competing interests exist.

of these fatalities [1]. Antimicrobial resistance poses a significant challenge in the management of infections caused by these bacterial species. In 2019, infectious diseases caused by antibiotic-resistant *S. aureus* and *S. pneumoniae* were responsible for 900,000 and 700,000 deaths, respectively. The burden of bacterial infectious diseases is particularly severe in low- and middle-income countries (LMIC) [2], underscoring the urgency to explore universally applicable hygiene control methods.

Although antibiotics are commonly used to treat infectious diseases, toxins produced by bacteria can cause diseases even after sterilization [3]. For example, *S. aureus* produces staphylococcal enterotoxins (SEs), which are significant contributors to the development of foodborne diseases [4]. In the European Union alone, 114 foodborne disease outbreaks in 2018 were attributed to SEs [5]. SEs cause symptoms such as nausea, vomiting, abdominal pain, cramps, and diarrhea, even at intakes of less than 1 μg [4]. Moreover, SEs are resistant to heat treatment and low pH [6], necessitating the development of novel approaches to treat and prevent diseases caused by bacteria and their toxins.

Ozone gas demonstrates potent antimicrobial activity against various micro-organisms, such as bacteria, viruses, fungi, yeast, and protozoa [7–9]. Ozone disrupts the integrity of bacterial cell envelope by oxidizing phospholipids and lipoproteins. In addition, ozone inhibits fungal cell growth. In the case of viruses, ozone has been reported to damage the viral capsid and disrupt the reproductive cycle by interfering with viral cell contact through peroxidation [7]. Additionally, ozone gas degrades fungal toxins such as mycotoxins [10]. However, high concentrations of ozone gas are harmful to humans. Exposure to more than 0.08 ppm of ozone gas for a few hours significantly decreases lung function in healthy adults [11]. Furthermore, in vitro exposure to ozone gas causes the death of human oral epithelial cells and gingival fibroblasts [12].

Ozonated water also exhibits antimicrobial activity. Because ozone volatilization from the surface of ozonated water is relatively low, ozonated water is less harmful than ozone gas. Previous studies on ozonated water primarily focused on assessing hand hygiene protocols in healthcare settings and the sterilizing food processing equipment [13–15]. However, ozone gas exhibits instability when dissolved in water, resulting in a relatively short half-life of less than 1 h [16, 17]. To address these challenges, the ultrafine bubble (UFB) technology has attracted considerable attention.

UFBs, with a diameter of less than 1 μm, demonstrate high stability in water [18]. In a previous study, we developed an ozone ultrafine bubble water (OUFBW) generator that exerted bactericidal activity against various bacterial species, including antibiotic-resistant strains [19]. However, the effects of OUFBW on bacterial toxins remain unclear. In this study, we aimed to determine whether OUFBW can degrade bacterial toxins and examine the effects of OUFBW treatment on various bacterial toxins.

## Materials and methods

### Production of OUFB-solution (OUFBW, OUFB-phosphate-buffered saline (OUFB-PBS), OUFB-Roswell Park Memorial Institute (OUB-RPMI)), and ultrafine bubble water (UFBW)

OUFBW was manufactured using oxygen present in air and distilled water (DW), as described in our previous study [19]. Briefly, ozone gas was generated using a dielectric barrier discharge ozone generator (10 g/h) (Futech-Niigata LLC, Niigata, Japan) with 90% of the oxygen supplied by an oxygen concentrator (flow rate: 1 L/min) (BMC Medical Co., Ltd, Beijing, China). The OUFBW was generated using a micro blender (Futech-Niigata LLC) and circulated in a stainless steel tank. UFBW was produced in the same manner as OUFBW using room air

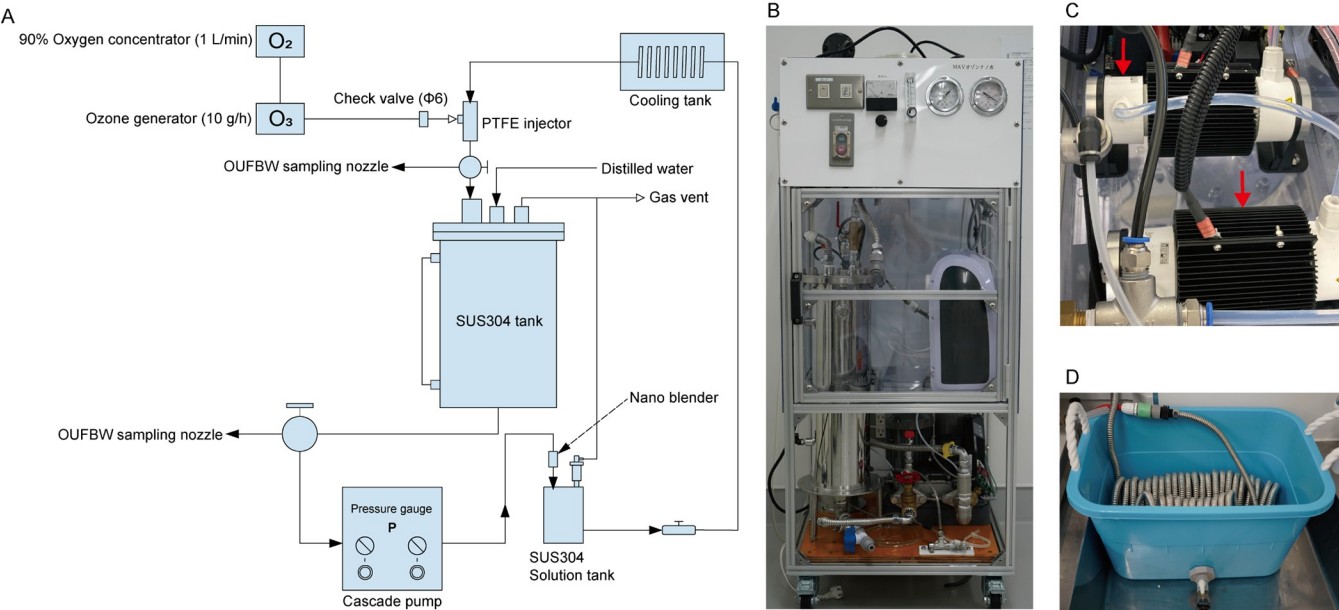

**Fig 1. Structure of ozone ultrafine bubble water (OUFBW) generator.** (A) Schematic of the OUFBW generator. (B) Overall image of the OUFBW generator. It can generate 4 L of OUFBW (3–5 ppm ozone concentration) and ultrafine bubbles (UFBW) in 15 min. (C) Red arrows indicate the dielectric barrier discharge ozone generators that generate ozone gas (10 g/h). (D) Filling the cooling tank with ice and water keeps the water temperature < 10˚C inside the OUFBW generator.

instead of ozone gas. The structure of the OUFBW generator is shown in Fig 1. To prepare OUFBW at various ozone concentrations, OUFBW containing 3–5 ppm ozone was serially diluted with DW. For the cytotoxicity assays, OUFB-PBS and OUFB-RPMI were prepared by diluting OUFBW with 10×PBS (Nacalai Tesque Inc., Kyoto, Japan) and 10×RPMI-1640 medium (Sigma-Aldrich, St. Louis, MO, USA), respectively. The ozone concentrations in the OUFB solutions were measured using a digital pack Test Ozone (Kyoritsu Chemical Check Lab, Tokyo, Japan).

## Silver staining assay and western blotting assay

Purified leukotoxin (LtxA) (3.6 μg), recombinant pneumolysin (rPLY) (2.2 μg), and purified staphylococcal enterotoxin A (SEA) (3.5 μg) were added to 500 μL of OUFBW or UFBW and incubated for 5 min. After that, the mixture was concentrated using Amicon Ultra-0.5 centrifugal filter devices (Merck KGaA, Darmstadt, Germany) and separated by standard sodium dodecyl-sulfate polyacrylamide gel electrophoresis (SDS-PAGE) using 12% acrylamide gel (Bio-Rad, Hercules, CA, USA). The resulting gels were stained with silver using the Pierce™ Silver Stain Kit (Thermo Fisher Scientific, Waltham, MA, USA).

## Isolation of human neutrophils and erythrocytes

Heparinized whole blood samples were obtained from three healthy donors (A, B and C) between April 17, 2023, and July 26, 2023. Neutrophils were isolated using Polymorphprep (Axis-Shield, Dundee, UK) in a 1:1 ratio of PBS, and centrifuged at 470×$g$ for 30 min. The layers containing neutrophils were carefully collected, and residual red blood cells were lysed using ACK Lysing Buffer (Lonza, Basel, Switzerland). Human neutrophils were counted using a Countess II FL (Thermo Fisher Scientific) and used for subsequent experiments. Erythrocytes were isolated by centrifugation at 180×$g$ for 10 min from whole blood and washed thrice

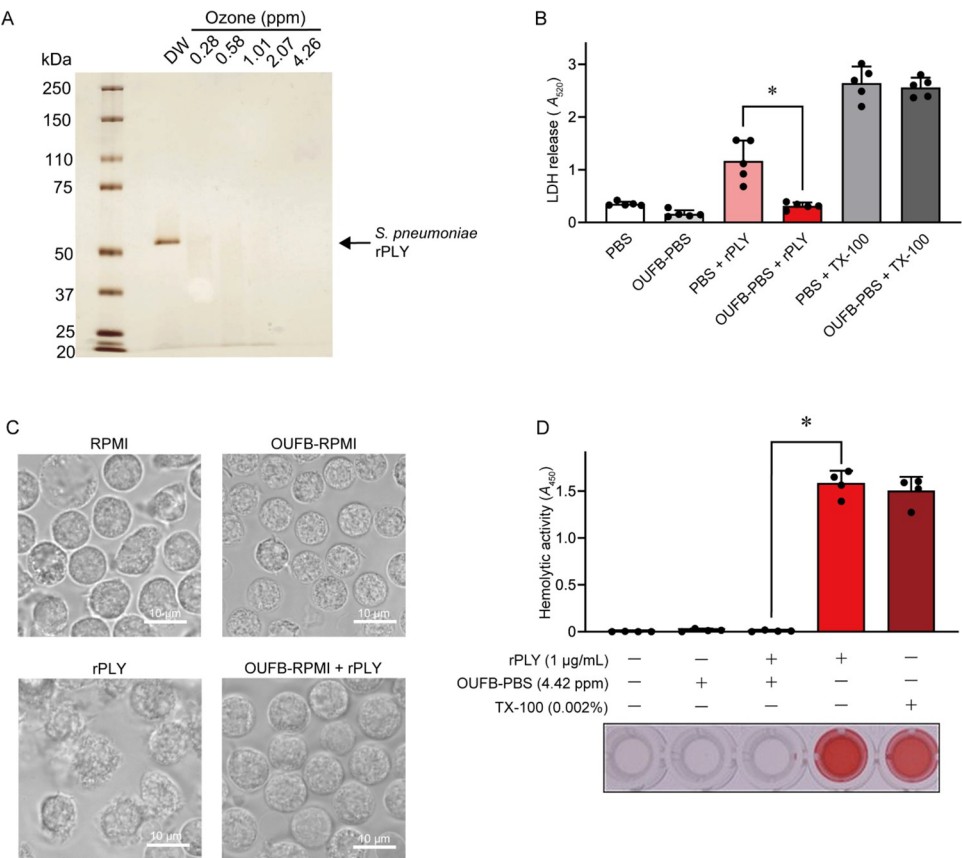

**Fig 2. Ozone ultrafine bubble (OUFB) degraded recombinant pneumolysin (rPLY).** (A) rPLY was added to OUFBW (0.28–4.26 ppm of ozone concentration) or distilled water, followed by SDS-PAGE and silver staining. (B) Human neutrophils ($1\times10^5$ cells/100 µL) were exposed to rPLY (10 pg/cells) in the presence or absence of OUFB-PBS containing 4.47 ppm of ozone for 3 h, followed by LDH assay. Data are presented as the mean ± SD of quintuplicate experiments and were evaluated using one-way analysis of variance with Dunnett's multiple-comparison test, * $P < 0.05$. N = 5 wells per each group (from one healthy donor). (C) Human neutrophils ($5\times10^5$ cells/100 µL) were exposed to rPLY (10 pg/cells) in the presence or absence of OUFB-RPMI (3.52 ppm ozone) for 1 h. Representative microscopic images were shown. (D) rPLY (final concentration 1 µg/mL) or Triton X-100 (TX-100) were added to human erythrocytes in PBS or OUFB-PBS (4.42 ppm of ozone concentration). Samples were incubated at 37˚C for 1 h, followed by a hemolytic assay. TX-100 was used as a positive control. Data are presented as the mean ± SD of quadruplicate experiments and were evaluated using a one-way analysis of Dunnett's multiple-comparison test, * $P < 0.05$. N = 4 wells per each group (from one health donor).

with PBS. All experimental procedures were approved by the Ethics Committee of Niigata University and were conducted following the approved guidelines. Written informed consent was obtained from all donors in the presence of a third party before participation in the study (permit # 2018–0075). Human blood was obtained from one donor per experiment (Fig 2B; N = 5 wells from donor A, Fig 2D; N = 4 wells from donor B, Fig 3B; N = 5 wells from donor C), and human cells were used for the experiment immediately after blood collection.

## Cytotoxicity assay and hemolytic assay

For cytotoxicity assay, 1 µL of LtxA (0.05 µg/mL) and 1 µL of rPLY (1 µg/mL) were added to 1 mL of PBS or OUFB-PBS containing 4.47 ppm of ozone followed by abolishment of OUFB by ultrasonication at 43 kHz for 5 min [18]. Human neutrophils were seeded onto a 96-well plate at a density of $1.0\times10^5$ cells/well in 100 µL of serum-free RPMI-1640 medium. After that, PBS,

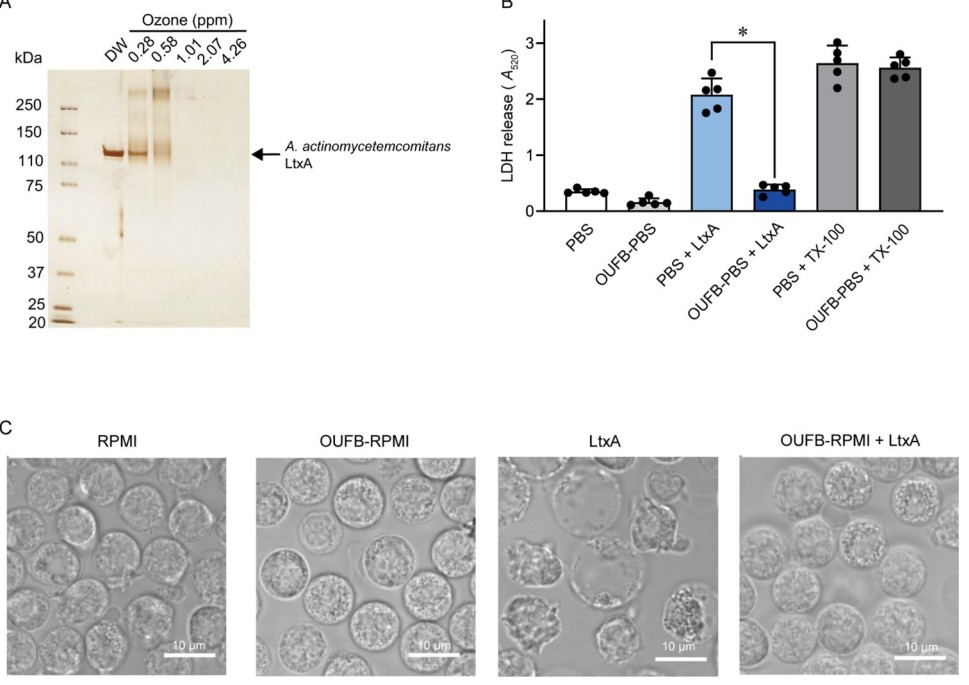

**Fig 3. Ozone ultrafine bubble (OUFB) degraded *Aggregatibacter actinomycetemcomitans* leukotoxin (LtxA) and abolish the cytotoxicity.** (A) LtxA was exposed to OUFBW (0.28–4.26 ppm of ozone concentration) or distilled water, followed by SDS-PAGE and silver staining. (B) Human neutrophils ($1\times10^5$ cells/100 μL) were exposed to LtxA (50 fg/cells) in the presence or absence of OUFB-PBS containing 4.47 ppm of ozone for 3 h, followed by lactate dehydrogenase (LDH) assay. Triton X-100 (TX-100) was used as a positive control. Data are presented as the mean ± SD of quintuplicate experiments and were evaluated using one-way analysis of variance with Dunnett's multiple-comparisons test, * $P < 0.05$. N = 5 wells per each group (from one healthy donor). (C) Human neutrophils ($5\times10^5$ cells/100 μL) were exposed to LtxA (50 fg/cell) in the presence or absence of OUFB-RPMI (3.52 ppm ozone) for 1 h. Representative microscopic images were shown.

OUFB-PBS, LtxA (final concentration: 50 fg/cell), rPLY (final concentration: 10 pg/cell), OUFB-treated LtxA, OUFB-treated rPLY, or 0.1% Triton X-100 was added to the culture and incubated at 37˚C in an atmosphere of 95% air and 5% $CO_2$ for up to 3 h followed by cytotoxicity analysis using a lactate dehydrogenase (LDH)-cytotoxicity test (Wako Pure Chemical Industries, Osaka, Japan) and optical microscopic analysis using BIOREVO BZ-9000 microscope (Keyence, Osaka, Japan). For hemolytic assay, 10 μL of rPLY (0.05 mg/mL) or 0.002% Triton X-100 was added to human erythrocytes (1% *v/v*) in 500 μL of PBS or OUFB-PBS containing 4.42 ppm of ozone. The samples were incubated at 37˚C for 30 min and centrifuged at 420×*g* for 10 min. The supernatant was pipetted into a 96-well plate, and hemolysis was measured using a Multiskan FC microplate photometer (Thermo Fisher Scientific, Waltham, MA, USA) at 450 nm.

## Bacterial culture and reagents

The methicillin-resistant *S. aureus* strain NILS2 and methicillin-susceptible strain NILS6 isolated from patients with severe *S. aureus* pneumonia [20] were cultured in tryptic soy broth (TSB; Becton Dickinson) for 48 h at 37˚C under aerobic conditions. Subsequently, the cultures were inoculated into fresh TSB and allowed to grow until they reached the exponential phase (optical density: 600 nm, 0.1). The bacteria were subsequently used for the bactericidal activity assays.

## Measurement of bactericidal activity of OUFBW

Bactericidal assays were performed as previously described [19]. Briefly, 1 μL of the bacterial cultures of *S. aureus* strains NILS2 and NILS6 with an optional density at 600 nm of 0.1 were added to 1 mL of OUFBW containing various concentrations (0.2–4 ppm) of ozone or UFBW for 1 min. After that, the cultures were diluted with DW and seeded onto TSB agar plates (Becton Dickinson). The agar plates were incubated under aerobic conditions at 37˚C for two days.

## Secreted alkaline phosphatase (SEAP) activity assay

HEK-Blue human Toll-like receptor (hTLR) 2 and hTLR4 cells (InvivoGen, San Diego, CA, USA) were cultured at 37˚C with 5% $CO_2$ in Dulbecco's modified Eagle's medium (Fujifilm Wako Pure Chemical Corporation), supplemented with 10% fetal bovine serum, 100 U/mL penicillin, 100 μg/mL streptomycin, 100 μg/mL Normocin (InvivoGen), and 100 μg/mL Zeocin (InvivoGen). These cell lines were suspended in HEK-Blue detection medium (InvivoGen), which contains a specific SEAP substrate that enables colorimetric detection of SEAP activity, and seeded at a density of $5\times10^4$ cells per 200 μL in 96-well plates. Then, 1 μL of Pam3CSK4 (1 mg/mL) or 1 μL of lipopolysaccharides (LPS) (0.5 mg/mL) from *Escherichia coli* strain O55:B5 (TLR4 ligand; Merck KGaA) was added to OUFBW or DW. After that, preprocessed Pam3CSK4 (final concentration, 2 ng/mL) and LPS (final concentration, 10 ng/mL) were added to HEK-Blue hTLR2 and hTLR4 cells, respectively, followed by incubation for 16 h. SEAP activity was measured at 620 nm using a Multiskan FC microplate photometer (Thermo Fisher Scientific).

## Statistical analysis

Data were analyzed by analysis of variance with Dunnett's multiple comparison test or Student's t-test using GraphPad Prism software version 9.4.0 (GraphPad Software, Inc., La Jolla, CA, USA).

## Results

### Degradation of rPLY by OUFB and inhibition of toxicity against human neutrophils and erythrocytes

Previous studies have indicated that PLY, a pneumococcal toxin, forms transmembrane pores through the cell membrane [21], inducing cell death and hemolysis in human neutrophils and erythrocytes, respectively [22, 23]. Therefore, we investigated the ability of OUFB to degrade rPLY and inhibit its cytotoxicity. rPLY was treated with OUFBW, followed by SDS-PAGE and silver staining. Following exposure to OUFBW containing > 0.28 ppm of ozone, the rPLY band almost disappeared (Fig 2A). However, the UFBW treatment did not affect the intensity of the rPLY bands (S1A Fig). Human neutrophils were exposed to rPLY in the presence or absence of OUFB-PBS for 3 h, followed by an LDH assay. Fig 2B illustrates a significant decrease in the release of LDH from human neutrophils following OUFB-PBS treatment compared to untreated cells (PBS + rPLY). Microscopic examination revealed that although treatment with rPLY appears damaged the cell membranes of neutrophils, OUFB-RPMI-treated rPLY did not induce any morphological changes in human neutrophils (Fig 2C). Additionally, OUFB-PBS significantly inhibited the hemolytic activity of rPLY (Fig 2D). These findings indicate that OUFB degrades rPLY, thereby reducing its toxicity.

## Degradation of LtxA by OUFB and inhibition of cytotoxicity against human neutrophils

LtxA, a toxin produced by *Aggregatibacter actinomycetemcomitans*, damages the human neutrophil membranes and causes cell death [24]. LtxA was treated with OUFBW or UFBW and subjected to SDS-PAGE and silver staining. Treatment of LtxA with OUFBW, containing > 1 ppm of ozone led to the near disappearance of the LtxA band (Fig 3A). In addition, S1B Fig shows that the band intensity was almost unchanged by the treatment of LtxA with UFBW. Therefore, we investigated the potential of OUFBW to eliminate LtxA toxicity. Human neutrophils were cultured with LtxA in the presence or absence of OUFB-PBS for 3 h, and subsequently subjected to LDH assay. Fig 3B shows a notable reduction in LDH release from human neutrophils following OUFB-PBS treatment compared to untreated cells (PBS + LtxA). Although LtxA appears damaged the neutrophil cell membranes, OUFB-RPMI-treated LtxA did not induce morphological changes in human neutrophils (Fig 3C). These data indicated that OUFB degraded LtxA and abolished its cytotoxicity.

## Bactericidal effects against *S. aureus* and degradation of SEA by OUFBW

*S. aureus* causes staphylococcal food poisoning by producing SEs in food [25]. We investigated the bactericidal effects of OUFBW and UFBW on *S. aureus* and determined their ability to degrade SEA, a type of SE that causes staphylococcal food poisoning. First, methicillin-resistant *S. aureus* strain NILS2 and methicillin-susceptible strain NILS6 were exposed to OUFBW containing 0.2–4 ppm of ozone or UFBW for 1 min. Fig 4A shows that exposure to approximately $\geq 0.8$ ppm OUFBW resulted in a > 99.9% decrease in the viability of both strains. On the other hand, the viability of *S. aureus* strains NILS2 and NILS6 hardly decreased upon exposure to UFBW (S2 Fig). Next, purified SEA was added to OUFBW and incubated for 5 min, followed by SDS-PAGE and silver staining. Fig 4B illustrates the complete disappearance of the SEA band after the OUFBW treatment, which strongly indicates the degradation of SEA by OUFBW. In the case of treatment with UFBW, the SEA band intensity remained almost unchanged (S1C Fig).

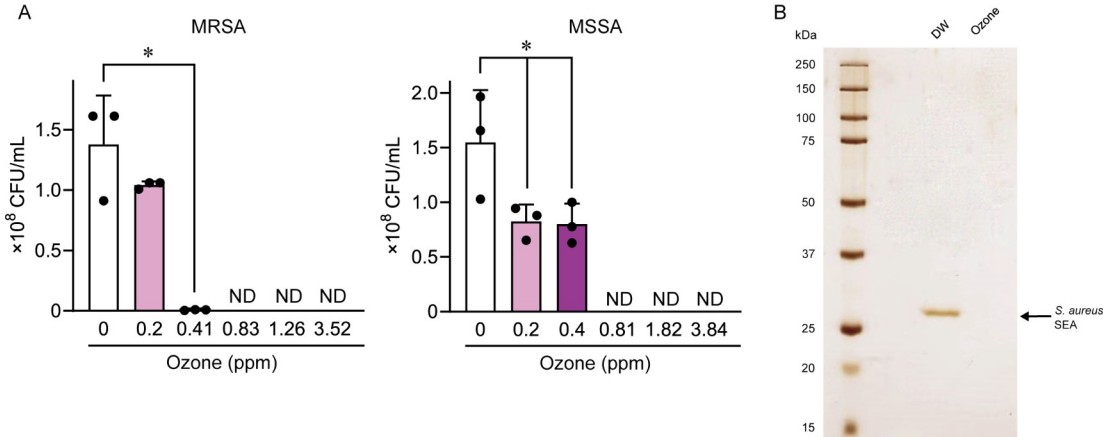

**Fig 4. OUFBW induced bactericidal effect against *Staphylococcus aureus* and decomposition of Staphylococcus enterotoxin A (SEA).** (A) Methicillin-resistant *S. aureus* strain (MRSA) NILS2 and methicillin-susceptible *S. aureus* strain (MSSA) NILS6 were exposed to OUFBW (0.2–4 ppm of ozone concentration) for 1 min. The bacterial load of *S. aureus* NILS2 and NILS6 was determined by colony count. Data are presented as the mean ± SD of triplicate experiments and were evaluated using a one-way analysis of variance with Dunnett's multiple-comparisons test. *$P < 0.05$ compared to 0 ppm (control) group. N = 3 for each bacterial strain. ND stands for undetected and indicates below the detection limit ($< 10^5$ CFU/mL). (B) Purified SEA was exposed to OUFBW (3.52 ppm of ozone concentration) or distilled water (DW), followed by SDS-PAGE and silver staining.

## Inhibition of Pam3CSK4-induced NF-κB activation by OUFBW in HEK 293 cells expressing hTLR2

Pathogen-associated molecular patterns derived from micro-organisms [26], such as lipoproteins, lipopeptides, and LPS, induce TLR-mediated proinflammatory cytokine production in host innate immune cells [27, 28]. Next, we examined the potential of OUFBW to inhibit the proinflammatory activities of Pam3CSK4, a synthetic triacylated lipopeptide that activates TLR2, and LPS, a major bacterial outer membrane component that activates TLR4. We used HEK-Blue cells expressing hTLR2 and hTLR4 as NF-κB receptor cells. HEK-Blue cells release SEAP into the cell culture medium in an NF-κB-dependent manner. Therefore, the activation of hTLR2 or hTLR4 was assessed by measuring SEAP activity. The OUFBW-treated Pam3CSK4 did not induce SEAP activity in HEK-Blue cells expressing hTLR2 (Fig 5A). Although treatment with LPS and OUFBW induced significantly lower SEAP activity in HEK-Blue cells expressing hTLR4 than in the DW + LPS group, the difference observed between the groups was relatively minor (Fig 5B).

## Discussion

Infectious and foodborne diseases are common global health concerns that emphasize the critical need for universally applicable and effective hygiene methods. Previous studies have highlighted the antimicrobial properties of ozone gas and ozonated water and demonstrated their efficacy in degrading fungal toxins [29]. Ozone, derived from oxygen, is attracting attention as a cost-effective disinfectant [30]. However, few studies have investigated the effects of OUFBW on bacterial toxins. Our findings indicated that OUFBW sterilizes *S. aureus*, including antibiotic-resistant strains, and degrades various bacterial protein toxins.

A previous study demonstrated a significant decrease in aflatoxin $B_1$, a potent hepatocarcinogenic fungal toxin, following exposure to ozonated water containing $1.7 \pm 0.17$ mg/L for 60 min [29]. In addition, Luo *et al.* suggested that the toxicity of the degradation products of aflatoxin $B_1$ produced by ozone significantly decreased [31]. Consistent with these findings, our

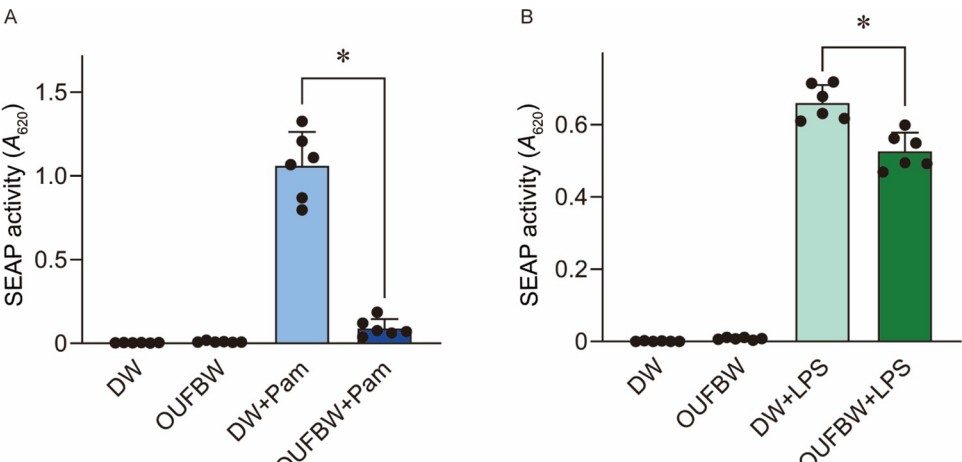

**Fig 5. Pam3CSK4 pretreated by ozone ultrafine bubble induces minimal TLR2 activation.** (A) Pam3CSK4 (Pam) or *Escherichia coli* LPS were added to ozone ultrafine bubble water (OUFBW) containing 4.21 ppm of ozone or distilled water (DW). After that, (A) HEK-Blue TLR-2 or (B) HEK-Blue TLR cells were stimulated with the Pam and the LPS, respectively. Secreted alkaline phosphatase (SEAP) levels were quantified using spectrophotometry at 620 nm. Data are presented as the mean ± SD of sextuplicate experiments and were evaluated using one-way analysis of variance with Tukey's multiple-comparison test, * $P < 0.05$. N = 6 for each experiment.

results indicated that OUFB treatment degraded rPLY, LtxA, and SEA within 5 min and abolished the cytotoxicity of rPLY and LtxA. Notably, SEA, a significant contributor to foodborne diseases, exhibits remarkable resistance to environments where *S. aureus* cannot thrive, such as high temperatures or the presence of proteolytic enzymes [32, 33]. These data underscore the potential of OUFBW as a valuable disinfectant in both the food industry and healthcare.

Bacteria induce inflammation in the human body through the release of pathogen-associated molecular patterns from bacterial cells [34]. Our study revealed a significant reduction in NF-κB activation by the synthetic bacterial lipopeptide Pam3CSK4, following treatment with OUFBW. However, the effect of OUFB treatment on LPS-induced NF-κB activation was minimal. Consistent with our findings, Noguchi *et al*. reported that therapy with ozonated water containing 2 ppm ozone partially suppressed the proinflammatory activity of LPS from *A. actinomycetemcomitans* and lipid A from *E. coli* [35]. LPS, a macromolecular glycolipid comprising a hydrophobic lipid A region attached to a long-branched carbohydrate chain, activates TLR4 [36, 37]. These findings suggest that ozone can degrade proteins and peptides such as bacterial toxins and Pam3CSK4, but not lipid A.

Although widely used disinfectants such as sodium hypochlorite, povidone-iodine, and ethanol demonstrate effective antimicrobial properties, they are also recognized for their cytotoxic effects on human cells [38–40] and pose a significant environmental burden [41, 42]. For example, although sodium hypochlorite can degrade a wide variety of biological molecules, such as proteins, amino acid peptides, lipids, and DNA [43], it is associated with substantial cytotoxicity in human cells [44]. Ozone is converted into oxygen and water immediately after oxidation by organic compounds [45, 46]. Consequently, ozone is considered harmless to living organisms and the environment after the reaction [10]. Previous in vitro studies have reported that OUFB solutions have low cytotoxicity toward various human cells, such as primary periodontal ligament fibroblasts and gingival epithelial cells [19, 47]. In this study, the OUFB solutions demonstrated minimal toxicity toward human neutrophils and erythrocytes. These results strongly imply that OUFBW is safer than the existing disinfectants.

Previous studies have indicated that ozone gas and ozonated water exert antimicrobial activities against various microorganisms such as bacteria, fungi, and viruses [48]. In the present study, OUFBW (0.8–4 ppm of ozone concentration) sterilized planktonic *S. aureus*, including antibiotic-resistant strains. However, in our previous study, OUFBW containing approximately 5 ppm ozone was unable to sterilize bacteria adhered to toothbrush and gauze completely [19]. Shichiri-Negoro *et al*. reported that treatment with OUFBW (9 or 11 ppm ozone) reduced the viable counts of *Candida albicans* in biofilms. However, the biofilms formed within 24 h were not completely removed [49]. These results indicate that OUFBW could not sterilize bacteria and fungi in a complicated structure. Microorganisms need to be directly exposed to OUFB to exert their antimicrobial properties.

The high burden of infectious diseases in LMICs is due to long-term health system deficiencies, compounded by poor living conditions, inadequate sanitation, limited access to health facilities and running water, overcrowding, and overpopulation [50]. Furthermore, LMICs are the most affected by foodborne diseases, with an estimated annual cost of US$ 110 billion in productivity losses, trade-related losses, and medical treatment expenses resulting from the consumption of unsafe food. This predicament is due to a lack of hygiene and sanitation facilities that are crucial for ensuring food safety [51]. The global commitment encapsulated in the Sustainable Development Goal 6 emphasizes the achievement of universal access to hygiene by 2030. Hence, to ensure accessibility, it is important to explore hygiene management methods that transcend material and financial constraints, ensuring accessibility for all [52]. OUFBW, which is produced from water and oxygen, is a safe and low-cost disinfectant that prevent infectious and foodborne diseases in LMIC.

## Supporting information

**S1 Fig. Ultrafine bubble water (UFBW) does not degrade bacterial toxins.** (A) Recombinant penumolysin, (B) *Aggregatibacter actinomycetemcomitans* leukotoxin, and (C) purified Staphylococcus enterotoxin A were added to UFBW or distilled water, followed by SDS-PAGE and silver staining.
(TIF)

**S2 Fig. Ultrafine bubble water (UFBW) does not induce a bactericidal effect against *Staphylococcus aureus*.** Methicillin-resistant *S. aureus* strain (MRSA) NILS2 and methicillin-susceptible *S. aureus* strain (MSSA) NILS6 were exposed to UFBW or distilled water (DW) for 1 min. The bacterial loads of *S. aureus* NILS2 and NILS6 were determined using colony counting. Data are presented as the mean ± SD of triplicate experiments and were evaluated using a one-way analysis of variance with a Student's t-test. $^*P < 0.05$. N = 3 for each bacterial strain. ND was undetected or below the detection limit ($< 10^5$ CFU/mL).
(TIF)

**S3 Fig. Original images of silver stain gels.** Unprocessed silver-stained images are shown in each figure. Lanes not induced in the final figure marked with an "X" above the lane. Images were obtained by scanning the gel using an image scanner.
(TIF)

**S1 Table. The minimal data set of each graph.**
(XLSX)

## Acknowledgments

We thank Mr. Tadashi Hiwatashi (Futech-Niigata LLC) for providing technical support.

## Author Contributions

**Conceptualization:** Yutaka Terao.

**Funding acquisition:** Fumio Takizawa, Hisanori Domon, Satoru Hirayama, Koichi Tabeta, Yutaka Terao.

**Investigation:** Fumio Takizawa, Satomi Tsutsuura.

**Methodology:** Fumio Takizawa, Hisanori Domon, Satoru Hirayama, Toshihito Isono, Karin Sasagawa, Daisuke Yonezawa, Akiomi Ushida, Satomi Tsutsuura.

**Project administration:** Hisanori Domon.

**Resources:** Toshihito Isono, Karin Sasagawa, Satomi Tsutsuura, Tomohiro Miyoshi, Hitomi Mimuro, Akihiro Yoshida.

**Supervision:** Hisanori Domon, Akiomi Ushida, Satomi Tsutsuura, Koichi Tabeta, Yutaka Terao.

**Visualization:** Satoru Hirayama.

**Writing – original draft:** Fumio Takizawa.

**Writing – review & editing:** Hisanori Domon, Yutaka Terao.

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
