## [Decision Letter · Decision Letter 0]

3 May 2024

PONE-D-24-12302Effective degradation of various bacterial toxins using ozone ultrafine bubble waterPLOS ONE

Dear Dr. Terao,

Thank you for submitting your manuscript to PLOS ONE. After careful consideration, we feel that it has merit but does not fully meet PLOS ONE’s publication criteria as it currently stands. Therefore, we invite you to submit a revised version of the manuscript that addresses the points raised during the review process.

We look forward to receiving your revised manuscript.

Kind regards,

Aijaz Ahmad, Ph.D.

Academic Editor

PLOS ONE

Journal Requirements:

Reviewers' comments:

Reviewer's Responses to Questions

**Comments to the Author**

1. Is the manuscript technically sound, and do the data support the conclusions?

Reviewer #1: Yes

Reviewer #2: Partly

2. Has the statistical analysis been performed appropriately and rigorously? 

Reviewer #1: Yes

Reviewer #2: Yes

3. Have the authors made all data underlying the findings in their manuscript fully available?

Reviewer #1: Yes

Reviewer #2: Yes

4. Is the manuscript presented in an intelligible fashion and written in standard English?

Reviewer #1: Yes

Reviewer #2: Yes

5. Review Comments to the Author

Reviewer #1: My comments to authors are as

1) Draft need writing improvement by native english speaker

2) In introduction section, author should mention how ozone acts antimicrobial i.e mechanism through which O3 kills microbes.

3) Author discuss and showed the anti-microbial effect of Ozone on bacterial cells, whether Ozone also have anti-viral properties should be discussed in draft

4) Author should also discuss or highlight the limitations which restricts use of ozone as antimicrobial agents

Reviewer #2: The manuscript “Effective degradation of various bacterial toxins using ozone ultrafine bubble water” has described the effect of Ozone Ultrafine Bubble Water (OUFBW) in the degradation of bacterial toxins and antibiotic resistant, non-resistant bacteria. Authors also have evaluated the effects on Human cells. The manuscript is well written and have strength in the proposed work. Authors have used appropriate experiments to validate their objectives. However, a few major concerns need to be addressed to improve the quality of the manuscript.

1. In all the experimental groups, a proper control is missing. Although, the authors have used Distilled Water or PBS as control group, however it is not an appropriate control with the OUFBW. A proper control would be the ‘Ultra fine bubble water (UFBW) or UFB-PBS which do not contain ozone and may contain sterile environmental air’. The ultrafine bubbles themselves may have significant physical and mechanical effects on bacteria or bacterial toxins which cannot be ignored. Authors are requested to use UFBW and OUFBW and repeat the experiment for Figure 2A, 3A and 4B i.e. degradation of rPLY, LtxA and SEA. It should justify the overall findings. Author can add the new result as supplementary or main figure and discuss it in main texts.

2. Authors have used 3 healthy donors blood samples and have performed the experiment on erythrocytes and neutrophils. In the given results, Figure 2B, 2D and 3B the N is different (N is 4 and 5). Authors have used biological replicates or technical replicates. Kindly clarify and mention in the legend.

3. Please mention N for all the sub-figures in the figure legend.

4. Figure 2C and 3C, authors have shown the damage to neutrophils ‘cell walls’. Please correct it with ‘cell membrane’ as the mammalian cells do not have cell wall. As per the description, this experiment probably was done in triplicates (3 healthy donors), authors can quantify the membrane damage and present it with statistical bar diagram. Authors may use Image J software and quantify the membrane damage by measuring the damaged portion of cell membrane. It should give numerical value which can be used to quantify, and it will be helpful for readers to easily understand the degree of membrane damage. However, authors can choose their own preference of quantifying this.

5. In the abstract section: line 30, it’s a repetition of sentence ‘high concentration of ozone gas’. Kindly correct it.

6. In material and method: line 111: please mention if the human cells were freshly/immediately used for the experiment after blood draw or they (3 donor cells) were frozen after blood draw and then used together for the experiment.

6. PLOS authors have the option to publish the peer review history of their article (what does this mean?). If published, this will include your full peer review and any attached files.

Reviewer #1: **Yes: **RAVINDER KUMAR

Reviewer #2: **Yes: **Rohit Patel

---

## [Author Response · Author response to Decision Letter 0]

6 Jun 2024

Response to Editor and Reviewers

Response to Editor 

We thank the editor for their critical comments that have helped us to improve our manuscript. As indicated in the responses below, we have considered all these comments and addressed each of them during our revision of the manuscript.

<Comment #1> Please ensure that your manuscript meets PLOS ONE's style requirements, including those for file naming.

<Response> According to the editor’s comment, we have ensured that our manuscript and figures adhere to PLOS ONE's style requirements and have renamed the file accordingly. 

<Comment #2> Please note that funding information should not appear in any section or other areas of your manuscript. We will only publish funding information present in the Funding Statement section of the online submission form. Please remove any funding-related text from the manuscript.

<Response> Following the editor’s comment, we have removed all funding-related text from the revised manuscript.

<Comment #3> We note that the grant information you provided in the ‘Funding Information’ and ‘Financial Disclosure’ sections do not match. When you resubmit, please ensure that you provide the correct grant numbers for the awards you received for your study in the ‘Funding Information’ section.

<Response> At the time of resubmission, the ‘Funding Information’, ‘Financial Disclosure’, and 'Grant numbers' have been revised to match their descriptions. We have described the revised Financial Disclosure statement in cover letter.

<Comment #4> Please confirm at this time whether or not your submission contains all raw data required to replicate the results of your study. 

<Response> Lines 503 and Table S1 (in the revised version): According to the editor’s suggestion, we submitted raw data files as the values used to build graphs for Figures 2B, 2C, 3B, 4A, 5A, 5B and Figure S2.

<Comment #5> Please include your full ethics statement in the ‘Methods’ section of your manuscript file. In your statement, please include the full name of the IRB or ethics committee who approved or waived your study, as well as whether or not you obtained informed written or verbal consent. If consent was waived for your study, please include this information in your statement as well.

<Response> Lines 120, 121-122 (in the revised version): Following the editor’s comment, we added a description of the full name of the IRB or ethics committee in line 120 (in the revised version). Additionally, we described written informed consent of this study in lines 121-122 (in the revised version) as follows: “Written informed consent was obtained from all donors in the presence of a third party before participating in the study (permit # 2018-0075).”

<Comment #6> Please ensure that your Figures adhere fully to journal guidelines and provide the original underlying images for all blot or gel data reported in your submission.

<Response> Lines 500-502 and S3 Figure (in the revised version): According to the editor’s comment and journal guidelines, we submitted raw images of silver stain gels of Figure 2A, 3A, 4B, Figure S1A, S1B, and S1C.

Response to Reviewer 1

We are grateful to Reviewer 1 for the critical comments and suggestions, which have helped us improve our paper considerably. As indicated in the following responses, we have considered all of these comments and suggestions in the revised version of our paper.

<Comment #1> Draft need writing improvement by native English speaker.

<Response> Based on the reviewer’s suggestion, the revised manuscript has been proofread by a native English speaker. The manuscript was revised according to the proofreading. The revised sections are highlighted. Additionally, we have submitted an English proofreading certificate for this manuscript.

<Comment #2> In introduction section, author should mention how ozone acts antimicrobial i.e mechanism through which O3 kills microbes.

<Comment #3> Author discuss and showed the anti-microbial effect of Ozone on bacterial cells, whether Ozone also have anti-viral properties should be discussed in draft

<Response to comment #2 and #3> Lines 62-65 (in the revised version): Following the reviewer comment, we added a description of the antimicrobial mechanism of ozone as follows: “Ozone disrupts the integrity of the bacterial cell envelope through oxidation of the phospholipids and lipoproteins. In addition, ozone inhibits fungal cell growth. In the case of viruses, ozone has been reported to damage the viral capsid and disrupt the reproductive cycle by interfering with viral cell contact through peroxidation.” 

The targets of this study were bacteria and bacterial toxins. Therefore, we did not analyze the effects of ozone ultrafine bubble water on the viruses.

<Comment #4> Author should also discuss or highlight the limitations which restricts use of ozone as antimicrobial agents

<Response> Lines 309-318 (in the revised version): According to the reviewer’s suggestion, we added a paragraph in the discussion section about the limitations that restrict the use of ozone as an antimicrobial agent in lines 309-318 as follows: “Previous studies indicated ozone gas and ozonated water exert antimicrobial activity against various microorganisms such as bacteria, fungi, and virus. In the present study, OUFBW (0.8–4 ppm of ozone concentration) sterilized planktonic S. aureus, including antibiotic-resistant strains. However, in our previous study, OUFBW containing approximately 5 ppm ozone was unable to completely sterilize bacteria adhered to toothbrushes and gauze. Shichiri-Negoro et al. reported that treatment with OUFBW (9 or 11 ppm ozone) reduced the viable counts of Candida albicans in biofilms. However, biofilms that formed within 24 h were not completely removed. These results indicate that OUFBW could not sterilize bacteria and fungi in a complicated structure. To exert antimicrobial properties, microorganisms need to be directly exposed to OUFB.”

With the addition of the description, we have added references and changed the reference order.

Response to Reviewer 2

We thank Reviewer 2 for their critical comments and suggestions, which have helped us improve our paper considerably. According to your comments, we have performed some additional experiments. And then, we added a supporting Figure in response to your comment. As indicated in the following responses, we have considered the comments and suggestions in the revised manuscript.

<Comment #1> In all the experimental groups, a proper control is missing. Authors are requested to use UFBW and OUFBW and repeat the experiment for Figure 2A, 3A and 4B i.e. degradation of rPLY, LtxA and SEA. It should justify the overall findings. Author can add the new result as supplementary or main Figure and discuss it in main texts.

<Response> Figures S1, S2 and lines 86, 92, 93, 101, 106, 150, 151, 169, 179, 180, 204, 205, 206-208, 228, 231, 233, 234, 237, 238 and 489-503 (in the revised version): We agree with the reviewer’s comments, and we performed additional experiments about whether ultrafine bubble water exerts bactericidal activity and degrade rPLY, LtxA and SEA. Ultrafine bubble water showed no bactericidal activity against Staphylococcus aureus and did not decompose bacterial toxins. Thank you for your comment. We have added Figure S1 and S2 to the Supporting Information. We have also added Figure legend for these Figures in lines 489-499 as follows: “S1 Fig. Ultrafine bubble water (UFBW) does not degrade bacterial toxins. (A) Recombinant penumolysin, (B) Aggregatibacter actinomycetemcomitans leukotoxin, and (C) purified Staphylococcus enterotoxin A were added to UFBW or distilled water, followed by SDS-PAGE and silver staining. S2 Fig. Ultrafine bubble water (UFBW) does not induce a bactericidal effect against Staphylococcus aureus. Methicillin-resistant S. aureus strain (MRSA) NILS2 and methicillin-susceptible S. aureus strain (MSSA) NILS6 were exposed to UFBW or distilled water (DW) for 1 min. The bacterial loads of S. aureus NILS2 and NILS6 were determined using colony counting. Data are presented as the mean ± SD of triplicate experiments and were evaluated using one-way analysis of variance with a Student’s t-test. *P < 0.05. N = 3 for each bacterial strain. ND stands for undetected and indicates below the detection limit (< 105 CFU/mL).”

With the addition of the supporting information, we have added descriptions to the Materials and Methods and Results sections of the revised manuscript in lines 86, 92, 93, 101, 106, 150, 151, 169, 179, 180, 204, 205, 206-208, 228, 231, 233, 234, 237, and 238.

<Comment #2> Authors have used 3 healthy donors blood samples and have performed the experiment on erythrocytes and neutrophils. In the given results, Figure 2B, 2D and 3B the N is different (N is 4 and 5).

<Comment #6> In material and method: line 111(in the revised version): please mention if the human cells were freshly/immediately used for the experiment after blood draw or they (3 donor cells) were frozen after blood draw and then used together for the experiment.

<Response to comments #2 and #6>　Lines 123 and 124 (in the revised version): According to the reviewer’s suggestion, we have added a description to the revised manuscript as follows: “Human blood was obtained from one donor per experiment and the human cells were used for the experiment immediately after blood draw.”

<Comment #3> Please mention N for all the sub-Figures in the Figure legend.

<Response>　Lines 194, 200, 222, 245, 269, and 498 (in the revised version): According to reviewer’s suggestion, we have added a description of N to the figure legend of the revised manuscript in lines 194, 200, 222, 245, and 269.

<Comment #4> Figure 2C and 3C, authors have shown the damage to neutrophils ‘cell walls’. Please correct it with ‘cell membrane’ as the mammalian cells do not have cell wall. Authors can quantify the membrane damage and present it with statistical bar diagram.

<Response> Lines 184 and 212 (in the revised version): According to the reviewer’s suggestion, we have corrected ‘cell walls’ with ‘cell membrane’ in lines 184 and 212 (in the revised version). Although we attempted to quantify neutrophil cell membrane damage by 7-AAD and cytocalcein violet 450 (ab176749, Abcam, Cambridge, UK) staining, this was difficult because the dead cells were almost completely destroyed and removed by washing. Representative microscopic images are shown in Figures 2C and 3C.

<Comment #5> In the abstract section: line 30, it’s a repetition of sentence ‘high concentration of ozone gas’. Kindly correct it.

<Response> The Abstract has been revised accordingly. We apologize for the error.

---

## [Decision Letter · Decision Letter 1]

21 Jun 2024

PONE-D-24-12302R1Effective degradation of various bacterial toxins using ozone ultrafine bubble waterPLOS ONE

Dear Dr. Terao,

Thank you for submitting your manuscript to PLOS ONE. After careful consideration, we feel that it has merit but does not fully meet PLOS ONE’s publication criteria as it currently stands. Therefore, we invite you to submit a revised version of the manuscript that addresses the points raised during the review process.

Please submit your revised manuscript by Aug 05 2024 11:59PM. If you will need more time than this to complete your revisions, please reply to this message or contact the journal office at plosone@plos.org. Please include the following items when submitting your revised manuscript:A rebuttal letter that responds to each point raised by the academic editor and reviewer(s). You should upload this letter as a separate file labeled 'Response to Reviewers'.A marked-up copy of your manuscript that highlights changes made to the original version. You should upload this as a separate file labeled 'Revised Manuscript with Track Changes'.An unmarked version of your revised paper without tracked changes. You should upload this as a separate file labeled 'Manuscript'.If applicable, we recommend that you deposit your laboratory protocols in protocols.io to enhance the reproducibility of your results. Protocols.io assigns your protocol its own identifier (DOI) so that it can be cited independently in the future. For instructions see: https://journals.plos.org/plosone/s/submission-guidelines#loc-laboratory-protocols. Additionally, PLOS ONE offers an option for publishing peer-reviewed Lab Protocol articles, which describe protocols hosted on protocols.io. Read more information on sharing protocols at https://plos.org/protocols?utm_medium=editorial-email&utm_source=authorletters&utm_campaign=protocols.

We look forward to receiving your revised manuscript.

Kind regards,

Aijaz Ahmad, Ph.D.

Academic Editor

PLOS ONE

Journal Requirements:

Reviewers' comments:

Reviewer's Responses to Questions

**Comments to the Author**

1. If the authors have adequately addressed your comments raised in a previous round of review and you feel that this manuscript is now acceptable for publication, you may indicate that here to bypass the “Comments to the Author” section, enter your conflict of interest statement in the “Confidential to Editor” section, and submit your "Accept" recommendation.

Reviewer #1: All comments have been addressed

Reviewer #2: (No Response)

Reviewer #3: All comments have been addressed

2. Is the manuscript technically sound, and do the data support the conclusions?

Reviewer #1: Yes

Reviewer #2: Yes

Reviewer #3: Yes

3. Has the statistical analysis been performed appropriately and rigorously? 

Reviewer #1: N/A

Reviewer #2: N/A

Reviewer #3: Yes

4. Have the authors made all data underlying the findings in their manuscript fully available?

Reviewer #1: Yes

Reviewer #2: Yes

Reviewer #3: Yes

5. Is the manuscript presented in an intelligible fashion and written in standard English?

Reviewer #1: Yes

Reviewer #2: Yes

Reviewer #3: Yes

6. Review Comments to the Author

Reviewer #1: After revision draft looks OK for acceptance and publication

Author responded to comments in sufficient details

Reviewer #2: The authors have revised the manuscript in the light of the reviewers’ comments. Most of the comments have been addressed in a satisfied manor and draft has been revised.

1. However, the second comment from the previous request was not addressed which is critically important to clarify the technicality of the experiment.

“Authors have used 3 healthy donors blood samples and have performed the experiment on erythrocytes and neutrophils. In the given results, Figure 2B, 2D and 3B the N is different (N is 4 and 5). Authors have used biological replicates or technical replicates. Kindly clarify and mention in the legend.”

This is important to clarify how N=3 (healthy donor) became N=4 or N=5 in the results.

2. Cell membrane damage quantification (line 184 and 203 in revised draft):

Without a clear marker or proper quantification, it is difficult to conclude the membrane damage only by microscopic observation. These cells may change their granularity and phenotype in response to the toxin and may look like having damaged membrane. Authors are advised to modify the statements as ‘appears damaged the cell membrane/ appears damage the human neutrophil’.

Reviewer #3: I believe all of the comments raised by the reviewers are adequately addressed by the authors. The manuscript in its current form shall be accepted.

7. PLOS authors have the option to publish the peer review history of their article (what does this mean?). If published, this will include your full peer review and any attached files.

Reviewer #1: **Yes: **RAVINDER KUMAR

Reviewer #2: **Yes: **Rohit Patel

Reviewer #3: **Yes: **MUHAMMAD ISHFAQ

---

## [Author Response · Author response to Decision Letter 1]

25 Jun 2024

Response to Editor and Reviewer #2

We thank the editor and all reviewers for their critical reviews. As indicated in the responses below, we have considered all your comments and addressed each of them during our R2-version of the manuscript.

<Reviewer #2 Comment #1> However, the second comment from the previous request was not addressed which is critically important to clarify the technicality of the experiment. “Authors have used 3 healthy donors blood samples and have performed the experiment on erythrocytes and neutrophils. In the given results, Figure 2B, 2D and 3B the N is different (N is 4 and 5). Authors have used biological replicates or technical replicates. Kindly clarify and mention in the legend.” This is important to clarify how N=3 (healthy donor) became N=4 or N=5 in the results.

<Response #1> Lines 113-114, 123-124, 196, 203, 225-226 in the R2-version: According to the reviewer’s comment, we have added a description to the revised manuscript as follows, respectively: “Lines 113-114: Heparinized whole blood samples were obtained from three healthy donors (A, B and C) between April 17, 2023, and July 26, 2023.”, “Lines 123-124: Human blood was obtained from one donor per experiment (Fig 2B; N = 5 wells from donor A, Fig 2D; N = 4 wells from donor B, Fig 3B; N = 5 wells from donor C), and human cells were used for the experiment immediately after blood collection.”, “Line 196 (Fig. 2B): N = 5 wells per each group (from one healthy donor). “, “Line 203 (Fig. 2D): N = 4 wells per each group (from one health donor).”, “Lines 225-226 (Fig. 3B): N = 5 wells per each group (from one healthy donor).”

<Reviewer #2 Comment #2> Cell membrane damage quantification (line 184 and 203 in revised draft): Without a clear marker or proper quantification, it is difficult to conclude the membrane damage only by microscopic observation. These cells may change their granularity and phenotype in response to the toxin and may look like having damaged membrane. Authors are advised to modify the statements as ‘appears damaged the cell membrane/ appears damage the human neutrophil’.

<Response #2> Lines 185-187 and 215-216 in the R2-version: In accordance with the reviewer #2’s comment, we have modified the text that was pointed out as follows: “Lines 185-187: Microscopic examination revealed that although treatment with rPLY appears damaged the cell membranes of neutrophils, OUFB-RPMI-treated rPLY did not induce any morphological changes in human neutrophils (Fig 2C).” and “Lines 215-216: Although LtxA appears damaged the neutrophil cell membranes, OUFB-RPMI-treated LtxA did not induce morphological changes in human neutrophils (Fig 3C).”

<Journal Requirements> Please review your reference list to ensure that it is complete and correct. If you have cited papers that have been retracted, please include the rationale for doing so in the manuscript text, or remove these references and replace them with relevant current references. Any changes to the reference list should be mentioned in the rebuttal letter that accompanies your revised manuscript. If you need to cite a retracted article, indicate the article’s retracted status in the References list and also include a citation and full reference for the retraction notice.

<Response> Lines 337-493 in the R2-version: In accordance with the 'Journal Requirements', we have reviewed and revised the reference list.

---

## [Editor Report · Decision Letter 2]

27 Jun 2024

Effective degradation of various bacterial toxins using ozone ultrafine bubble water

PONE-D-24-12302R2

Dear Dr. Yutaka Terao

We’re pleased to inform you that your manuscript has been judged scientifically suitable for publication and will be formally accepted for publication once it meets all outstanding technical requirements.

Kind regards,

Aijaz Ahmad, Ph.D.

Academic Editor

PLOS ONE

---

## [Editor Report · Acceptance letter]

1 Jul 2024

PONE-D-24-12302R2 

PLOS ONE

Dear Dr. Terao, 

I'm pleased to inform you that your manuscript has been deemed suitable for publication in PLOS ONE. Congratulations! Your manuscript is now being handed over to our production team.

Kind regards, 

on behalf of

Dr. Aijaz Ahmad 

Academic Editor

PLOS ONE